# The protocol for a pilot feasibility trial of Improving Neurodevelopmental ouTcomes After prenatal Cannabinoid in uTero exposure (INTACT) study for a multi-center trial

Jessie R. Maxwell[1,2]*, Leigh-Anne Cioffredi[3,4], Maria M. Talavera-Barber[5,6], Matthew Henry[7], Sandra Beauman[1], Anne Hittson[1], Meggie McCoy[5], Laurie Chassereau[3], Jing Jin[8], Preetha A. Abraham[9], Linda Y. Fu[9], Hengameh Raissy[1], Jessica N. Snowden[10]

**1** Department of Pediatrics, University of New Mexico, Albuquerque, New Mexico, United States of America, **2** Department of Neurosciences, University of New Mexico, Albuquerque, New Mexico, United States of America, **3** Department of Pediatrics, University of Vermont Larner College of Medicine, Burlington, Vermont, United States of America, **4** Vermont Children's Hospital, Burlington, Vermont, United States of America, **5** Avera Research Institute, Avera McKennan Hospital & University Health Center, Sioux Falls, South Dakota, United States of America, **6** Department of Pediatrics, University of South Dakota Sanford School of Medicine, Sioux Falls, South Dakota, United States of America, **7** ECHO ISPCTN Data Coordinating and Operations Center, University of Arkansas for Medical Sciences, Little Rock, Arkansas, United States of America, **8** Department of Biostatistics, University of Arkansas for Medical Sciences, Little Rock, Arkansas, United States of America, **9** Environmental Influences on Child, Health Outcomes (ECHO) Program Institutional Development Award (IDeA) States Pediatric Clinical Trials Network, NIH, Rockville, Maryland, United States of America, **10** Department of Pediatrics, University of Arkansas for Medical Sciences, Little Rock, Arkansas, United States of America

* jrmaxwell@salud.unm.edu

## Abstract

### Background

Legalization of recreational cannabis use is expanding across the United States, and prenatal cannabis has steadily increased. Evidence suggests that many pregnant individuals use cannabis to relieve symptoms like nausea. Research has demonstrated an association between prenatal cannabinoid exposure and infant deficits in attention, planning, and memory. In other high-risk populations, interventions aimed at increasing parental responsiveness have improved cognitive functioning in the children. This pilot trial aims to utilize a contingent responding training program in birthing parent-infant dyads with prenatal cannabinoid exposure to assess the feasibility of recruitment, completion of the proposed intervention and adherence.

### Methods

This study will enroll post-partum birthing parents who used cannabinoid products during pregnancy at three clinical sites. After consenting and confirming eligibility, birthing parents will be oriented to the online program Play and Learning Strategies (ePALS) by the study team member, after which they will complete asynchronous monthly modules for 12 months that highlight aspects of contingent

purpose. The work is made available under the Creative Commons CC0 public domain dedication.

**Data availability statement:** Deidentified research data will be made publicly available when the study is completed and published.

**Funding:** The Environmental influences on Child Health Outcomes (ECHO) Program, Office of the Director, National Institutes of Health, supported this research under the award numbers U24OD024957, UG1OD024947, UG1OD024955, and UG1OD030019. As this study is being conducted under cooperative agreement funding mechanisms LYF and PA, employees of the National Institutes of Health, participated in study design, preparation of the manuscript and the decision to publish. The content is solely the responsibility of the authors and does not represent the official views of the National Institutes of Health.

**Competing interests:** The authors have declared that no competing interests exist.

**Abbreviations :** ABCD, Adolescent Brain Cognitive Development Study; ACOG, American College of Obstetricians and Gynecologists; AGA, appropriate for gestational age; ASAM, American Society of Addiction Medicine; Bayley-IV, Bayley Scales of Infant and Toddler Development Screening test, 4th Edition; $CB_1$, primary brain cannabinoid receptor; CoC, Certificate of Confidentiality; COVID-19; DSMB, Data Safety Monitoring Board; ECHO, Environmental influences on Child Health Outcomes; ECS, endocannabinoid system; ELEAT, Early Life Exposures Assessment Tool; EMR, electronic medical record; ePALS, Remote Play and Learning Strategies; Gen R, Generation R; HIPAA, Health Insurance Portability and Accountability Act; INTACT, Improving Neurodevelopmental ouTcomes After prenatal Cannabinoid in uTero exposure; ISPCTN, IDeA States Pediatric Clinical Trials Network; LEAF, Lifestyle and Early Achievement in Families Study; MHPCD, Maternal Health Practices and Child Development; MoBa, Norwegian Mothers and Child Cohort Study; MORE, Marijuana Opportunity Reinvestment and Expungement Act; NICU, Newborn Intensive Care Unit; NIH, National Institutes of Health; OPPS, Ottawa Prenatal Prospective Study; PALS, Play and Learning Strategies; RCT, Randomized Controlled Trial; RUCA, Rural-Urban Community Area; SGA, small for gestational age; THC, $\Delta^9$-tetrahydrocannabinol; WHO, World Health Organization

responding. Study staff at each site will be trained as coaches, meeting monthly with the birthing parent to review and reinforce the areas of focus. The primary objectives of the study will focus on the ability to recruit eligible birthing caregivers with cannabinoid use during pregnancy, the ability to retain participants for the intervention duration as measured through completion of the study session when the child is 12 months of age, and to assess the overall participant adherence of monthly sessions.

## Discussion

As cannabinoid use during pregnancy becomes more prevalent, it is critical that we can provide interventions to optimize infant developmental outcomes. This pilot trial is focused on adapting a proven intervention used in other high-risk populations to determine if it can be applied to this population. If successful, a future trial would focus on the efficacy of this intervention following prenatal cannabinoid exposure.

## Trial registration

Clinicaltrials.gov NCT06423664 .

---

## Introduction/Background

### Public health impact

Cannabinoids represent the third most frequently used substance during pregnancy, behind alcohol and tobacco [1]. Additionally, the recent shift to legalize the recreational use of cannabinoids in the United States (US) has been coupled with an increase of use during pregnancy [2]. Currently, recreational cannabinoid use is legal in 24 US states, two territories, and the District of Columbia (see Fig 1), and the Marijuana Opportunity Reinvestment and Expungement (MORE) Act, effectively ending the federal ban on cannabinoids, was passed by the House of Representatives on April 1, 2022 [3]. Significant social changes in cannabinoid use have been reported amidst this legal shift. According to the 2019 National Household Survey on Drug Use and Health, 5.4% of pregnant individuals in the US used cannabinoids in the past month, 28% higher than use just ten years earlier [4]. Furthermore, recent studies indicate rates of daily use and the quantity used during pregnancy are higher than previously established by cohort studies that enrolled in the early 1980s [5].

Of concern, the more widespread and frequent use of cannabinoids during pregnancy is occurring as concentrations of $\Delta^9$-tetrahydrocannabinol (THC), the psychoactive ingredient in marijuana, are also increasing in cannabinoid products. Eloshly et al. reported a 300% increase in the concentration of THC in recreational cannabinoid products since the 1990s [6,7]. This increase in potency, quantity, and prevalence of cannabinoid use in pregnancy suggests there is a substantial increase in fetal exposure to THC.

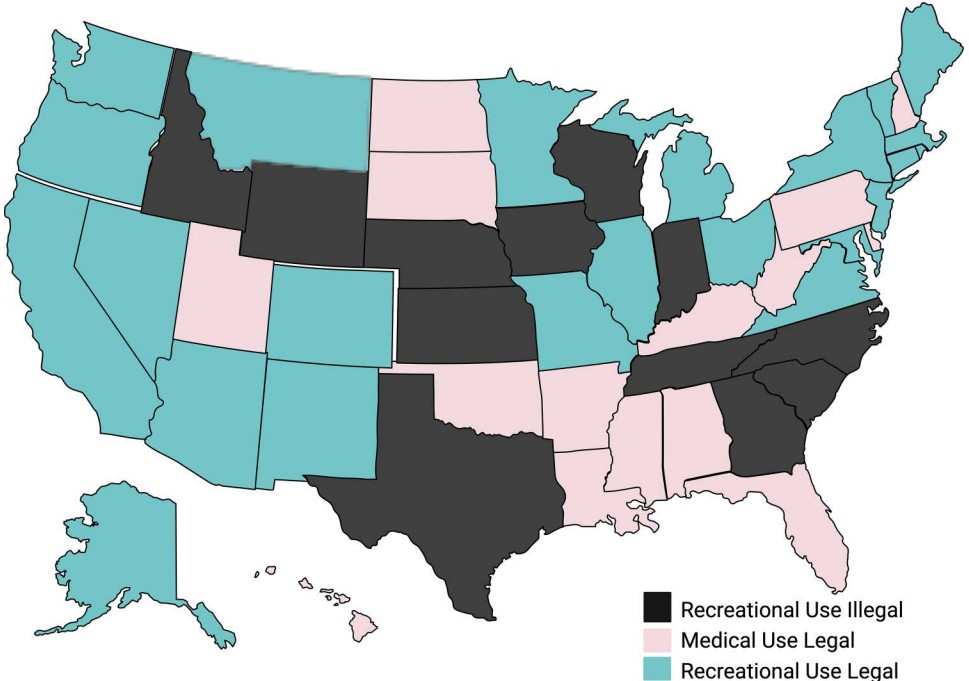

**Fig 1. Legalization status of cannabinoid in ISPCTN sites.** Created in BioRender. Maxwell, J. (2025) https://BioRender.com/f59k986.

## Individual impact

The potential for neurodevelopmental impacts due to intrauterine THC exposure is high. THC is a highly lipophilic molecule that readily crosses the placenta and has been shown to accumulate in fetal and placental tissue [8,9]. Moreover, the endogenous endocannabinoid system (ECS) is known to play a critical role in early brain development, in particular affecting proteins involved in axonal growth and neuronal connectivity [10–12]. Evidence shows the primary brain cannabinoid receptor ($CB_1$) is most intensely expressed in the human fetal brain in the mesocorticolimbic system, with particular predominance in the amygdala and hippocampus. In fact, expression of $CB_1$ is highest during gestation and drops precipitously after birth, indicating mid-gestation may be the developmental period of highest susceptibility to exposure to THC [13]. This growing body of evidence is complemented by emerging evidence that fetal neurotransmitter activity and neuronal connectivity are altered in those with prenatal cannabinoid exposure. Specifically, Wang et al. reported altered dopamine and opioid receptors in mid-gestation fetuses with prenatal cannabinoid exposure [14,15]. Additionally, Thomason et al. described fetal functional magnetic resonance imaging (fMRI) findings that suggested altered hippocampal connectivity in fetuses with prenatal cannabinoid exposure [16].

## Bridging the current knowledge gap

The growing body of evidence indicates the need for more studies investigating the effects of intrauterine THC exposure on infant developmental outcomes. Extant literature evaluating the impact of prenatal cannabinoid exposure on development during infancy includes three prospective longitudinal studies: the Ottawa Prenatal Prospective Study (OPPS), the Maternal Health Practices and Child Development Study (MHPCD), and the Generation R study [17–19]. Importantly, two of these enrolled participants during the late 1970s and early 1980s when exposures were vastly different than today. These studies are complemented by three other recent longitudinal cohorts, the Adolescent Brain Cognitive Development Study (ABCD),

Lifestyle and Early Achievement in Families Study (LEAF), and the Norwegian Mothers and Child Cohort Study (MoBa), which investigate the effects of prenatal cannabinoid exposure on neurodevelopment in childhood, but do not report developmental outcomes in the first 1–2 years of life. A recent systemic review and meta-analysis that included these longitudinal studies found overwhelming evidence of detrimental neonatal outcomes from prenatal cannabinoid exposure that included (1) low birth weight <2500g, (2) preterm birth, (3) Newborn Intensive Care Unit (NICU) admission, (4) small for gestational age (SGA) status, (5) low Apgar score at 1 minute of life and (6) small head circumference; all of which reflect abnormal fetal development [20,21]. Based on SGA status alone, a recent meta-analysis found poorer cognitive function during the first 12 years of life in children born SGA compared to appropriate for gestational age (AGA) matched for gestational age children [22].

In addition to differences in birth outcomes, the OPPS study found that intrauterine exposure to cannabinoids was highly associated with an increase in exaggerated startle reflex and tremors with a significantly diminished responsiveness to light in the neonatal period [17]. Altered sleep patterns were found in the MHPCD study with a trend toward increased irritability in those infants with prenatal cannabinoid exposure [18]. The MHPCD cohort also demonstrated higher amounts of cannabinoid use (>1 joint/day) during the 3rd trimester, which was associated with decreased mental scores on the Bayley Scales of Infant Development Scale at 9 months of age, a difference that disappeared at 18 months age [23]. In contrast, the Generation R study found evidence of increased aggression and inattention in 18-month-old girls [24]. These longitudinal studies also found executive functions altered by intrauterine exposure to cannabinoids, including attention, planning, vocalization, and working memory [25,26]. Altogether, these cognitive and behavioral functions are critically important for problem-solving situations that require integration and manipulation of basic visuoperceptual skills (see Fig 2) [27]. Thus, there is significant evidence to suggest that prenatal cannabinoid exposure has deleterious effects on the developing brain and significant potential to directly impact infant neurodevelopment.

## Birthing parent responsiveness and infant neurodevelopment

Intrauterine exposure to cannabinoids is only one exposure within a larger number of risk factors associated with altered infant neurodevelopment. Throughout gestation, maternal environmental factors (e.g., maternal depression,

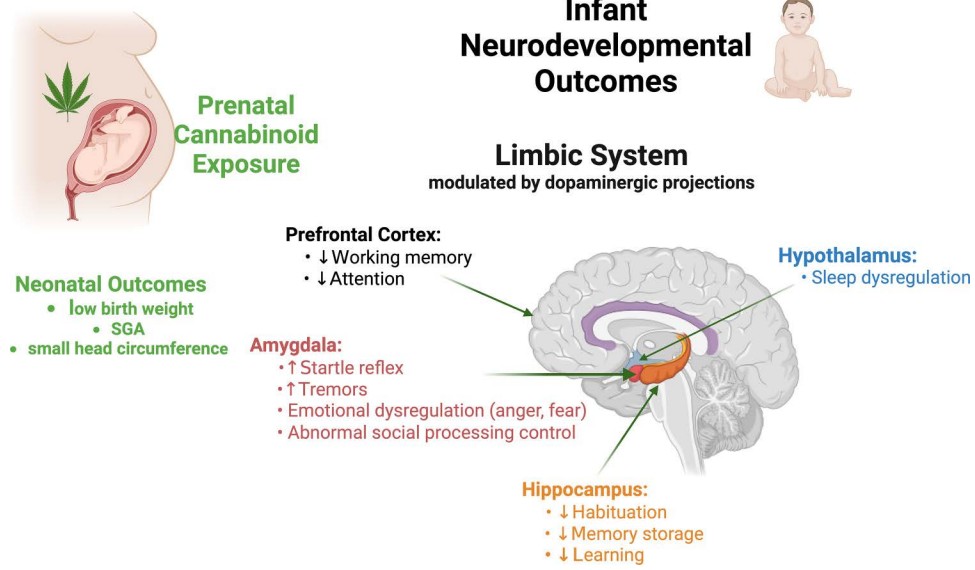

**Fig 2. Summary of the impact on neurodevelopment following prenatal cannabinoid exposure.** Created in BioRender. Barber, M. (2025) https://BioRender.com/q89g457.

smoking, nutrition) influence the cellular maturation and differentiation as well as the epigenome of the fetus resulting in exposure-influenced phenotypes [28]. After delivery, the early childhood environment is associated with differences in infant development [29]. A wealth of research on child development has established the strongly beneficial effects of adaptive and healthy parenting styles on normative development throughout all stages of childhood, with infancy being one of the most influential periods [29]. One of the strongest and most consistent predictors of infant neurodevelopmental outcomes is the quality of interactions infants have with their birthing parents. Birthing parent responsiveness, defined as timely, sensitive reactions to infant cues [30], has been repeatedly associated with positive language, social-emotional, and cognitive development in the infant [31–34].

In addition to the potential neurobiological damage that infants may experience following intrauterine exposure to cannabinoids, they are also more likely to be exposed to harsh and less sensitive parenting [35,36]. In fact, some hypothesize that it is precisely the presence of postnatal stressors that act as the "second hit" following prenatal cannabinoid exposure that precipitates meaningful changes in development [37]. Results supporting this hypothesis have been reported by Eiden and colleagues [29,36,38], who conducted a prospective observational study involving 97 mother-child dyads with prenatal cannabinoid and tobacco co-exposure and 69 unexposed dyads. In both groups, mother-child interactions were observed during infancy and evaluated for levels of maternal harshness and sensitivity through observation of warmth and positive affect. In addition, mothers completed measures of anger and depression, and children completed tasks at 2 years of age measuring autonomic and emotion regulation. Finally, maternal reports of child behavior problems were assessed at age 2 years. Results showed that mothers of cannabinoid-tobacco-exposed infants reported significantly higher levels of anger and depression and exhibited significantly higher levels of harshness and lower levels of sensitivity during mother-infant interactions compared to mothers of unexposed infants. In turn, levels of maternal harshness during mother-infant interaction were inversely associated with toddler autonomic regulation, and levels of maternal sensitivity were positively associated with toddler emotion regulation. Finally, maternal sensitivity during a child's infancy was inversely associated with child behavior problems at age 2 years. The findings of these important studies suggest that mothers who use cannabinoids prenatally may be more likely to exhibit maladaptive interaction patterns with their infants characterized by overly harsh and insensitive responses to infant behavioral cues. Such interaction patterns are predictive of subsequent deficits in toddler self-regulation and behavioral problems.

## PALS as a potential intervention

This pilot trial proposes to implement the evidence-based Play and Learning Strategies (PALS) program as an experimental intervention to enhance the cannabinoid-exposed infant's post-natal environment and development by systematically strengthening caregiver (i.e., birthing parent) sensitivity and responsiveness. Although yet untested for caregivers of infants exposed to cannabinoids prenatally, PALS is a rigorously tested intervention in other populations [39,40] that has been shown to strengthen caregiver responsiveness and sensitivity. Sensitive parenting has been associated with decreased rates of child maltreatment and improved infant social-emotional behavior and developmental outcomes, suggesting there may be broad benefits from this intervention [31,39–42]. Additionally, the program has demonstrated significant increases in multiple responsiveness behaviors that facilitate infants' growth in social-emotional, communication, and cognitive competence. Specifically, PALS consists of intervention sessions in which a trained coach uses the curriculum in a flexible manner to meet the individual birthing parent's learning needs. Each PALS session is up to one hour in duration and includes scripted didactic, probing questions, video segment sharing to illustrate each concept, and points for discussion [39,42]. The PALS curriculum has since been adapted to a virtual platform (i.e., ePALS), which will be implemented in this pilot trial [43].

Historically, the PALS/ePALS intervention has been utilized among mothers whose infants were born at very low birth weights, mothers with depressive symptoms, mothers from low-income families, mothers whose infants were born extremely premature, and socially disadvantaged mothers [31,39–42]. Results have included increased word use in the

infant, an increase in attention in the infant, and an increase in positive behavior during the birthing parent-child interaction by both participants. Evidence suggests the PALS/ePALS program impacts children's behaviors through a change in birthing parent responsiveness behavior and improved infant social and emotional skills. Birthing parents who have participated in PALS/ePALS have shown an increase in verbal scaffolding (i.e., the ability to add tailored support to the infant to aid in learning) and their children have greater gains in cognitive scores as compared to birthing parents and infants who did not receive the intervention [31,39–42]. With the recent COVID-19 pandemic, interventions requiring in-home visitation have either not occurred or been significantly reduced. Thus, this protocol proposes using the ePALS intervention, which has expanded the reach of the evidence-based PALS intervention by utilizing technology (i.e., video conferencing and online learning management platforms) to promote social-emotional development in early childhood [43].

### Establishing a gold standard for contingent response training

Currently, there is no gold standard for assessing the effect of contingent response training for birthing parents on neuro-developmental outcomes for infants younger than 2 years of age. This gap in knowledge is concerning due to this crucial time for a child's cognitive and social development.

Collectively, findings from the current research underscore a myriad of developmental challenges facing infants with prenatal cannabinoid exposure stemming from possible in utero neurologic insults, as well as potential ongoing environmental threats stemming from suboptimal birthing parent-child interactions. Evidence suggests that early intervention focusing on improving caregiving skills among birthing parents of infants prenatally exposed to cannabinoids may help to offset or counteract delayed infant neurodevelopment [44,45]. Among birthing parents with depression from low-income settings, training interventions have been shown to improve birthing parent skills and behavior, birthing parent-infant interaction quality, and infant developmental outcomes [43,46–48]. Thus, this pilot feasibility trial will focus on birthing parent interventions modeled on PALS/ePALS with the goal of improving the infant developmental outcomes of this highly vulnerable population. The Environmental influences on Child Health Outcomes IDeA States Pediatric Clinical Trials Network (ECHO ISPCTN) ensures that research trials are accessible to children in rural or underserved regions and works to improve research capacity in states with historically lower national funding. The ECHO ISPCTN has experience in conducting clinical trials in high-risk populations and will utilize this expertise to ensure similar success in this trial.

## Methods/design

### Study design and setting

Improving Neurodevelopmental ouTcomes After prenatal Cannabinoid in uTero exposure (INTACT) will be a 12-month (each month defined as 28 days), multi-site pilot feasibility trial that will assess the ePALS curriculum in birthing parents with cannabinoid use during pregnancy. The trial will examine 1) recruitment rates, 2) curriculum completion rates when the infant is 12 months of age, and 3) completion rates of individual INTACT intervention coaching sessions. The INTACT intervention is adapted from the ePALS curriculum and consists of remote monthly online educational videos followed by monthly meetings with coaches. The duration of the study for each birthing parent-infant dyad is 12 months, with all interactions with study staff (i.e., check-ins, monthly coaching sessions) following the delivery hospitalization being remote. In the development of this protocol, we have included three ECHO ISPCTN sites in states with different legalization statuses of cannabinoid use to ensure all aspects of cannabinoid legality that could affect study enrollment and participation were considered. The sites selected will include New Mexico (recreational cannabinoid use legalized in 2021 and effective April 2022), Vermont (recreational cannabinoid use legalized in 2018), and South Dakota (recreational cannabinoid use remains illegal).

This study protocol has been approved by the University of Arkansas for Medical Sciences Institutional Review Board (IRB), study number FWA00001119. The University of Arkansas for Medical Sciences (UAMS) IRB is duly constituted, fulfilling all requirements for diversity, and has written procedures for initial and continuing review of human subjects'

research; prepares written minutes of convened meetings; and retains records pertaining to the review and approval process. The UAMS IRB is organized and operates in compliance with DHHS regulations as described in 45 CFR part 46 (i.e., The Common Rule) and, after January 20, 2019, the Revised Common Rule and with FDA regulations as described in 21 CFR Parts 50 and 56.

## Justification of study design

This is an open-label study with all participants completing the ePALS intervention. This approach is being taken to allow for feasibility testing, including the feasibility of delivering the INTACT intervention with fidelity, so that a larger-scale trial can be designed upon completion of this study. Once this study has established feasibility, a larger-scale trial would include a control group. As this is a short (12 months of birthing parent participation) feasibility pilot trial, we intentionally are not collecting clinical outcomes; the future randomized controlled trial (RCT) would include a post-intervention assessment of birthing parent/infant interaction quality and assessment of infant neurodevelopment at 2 years of age via administration of the Bayley Scales of Infant and Toddler Development Screening Test, 4th Edition (Bayley-IV) [49].

As this pilot trial will examine the feasibility of implementing this intervention as well as the feasibility of recruitment and study completion, it will provide formative data that will be used to design a future fully powered RCT to test the efficacy of the INTACT intervention within a population with cannabinoid exposure during pregnancy. This study will address the following key elements that are integral to inform a larger RCT:

1) Feasibility of recruiting from this vulnerable, high-risk population and estimations of recruitment rate.

2) Participants should be able to complete the coaching session when the infant is 12 months of age.

3) Participants should be able to sufficiently engage in the intervention with appropriate adherence.

Additionally, the choice to implement the remotely delivered ePALS curriculum provides a means to reach a more remote and rural population that might be otherwise excluded. Completion of this trial on the remote platform allows this high-risk population access to additional interventions and can pave the way for the success of future remote interventions in this population.

## Study population with eligibility criteria

We anticipate 20 birthing parent-infant dyads will enroll, determined to be eligible to participate in the intervention, and complete at least the first coaching session across three sites (i.e., ~6–7 dyads per site that meet all inclusion-exclusion criteria.) Consent will only occur after the infant is born to ensure the preliminary eligibility (i.e., inclusion-exclusion). Final eligibility to participate in the intervention will be determined (following obtaining informed consent) via the Cannabinoid Use Survey, which assesses and excludes opioid and other illicit drug use during pregnancy. As the goal of this study is to inform an eventual RCT to test an intervention designed to improve the outcome of infants with prenatal cannabinoid exposure, we are excluding birthing parents with multi-substance use to minimize confounding factors, except tobacco use. Following the completion of the Cannabinoid Use Survey, we expect a screen-fail rate of 60%.

Of note, tobacco use will not be an exclusion criterion due to the frequency of concurrent use with cannabinoids in the proposed study population. Given the voluntary nature of the study and that intervention eligibility relies on self-reported cannabinoid use, some potentially eligible individuals may not self-identify, which could limit the number of intervention participants. This aspect is one of the feasibility components that will be assessed with this pilot trial.

**Inclusion criteria for the birthing parent.**

• Age of majority, as defined by the state of residency

• Cannabinoid use during pregnancy as determined by self-report

- Ability to speak, read, and understand English

- Childbirth at one of the hospitals where study team members have clinical privileges to access medical records

- Parental custody of the infant

- Singleton pregnancy with live birth

- In possession of an electronic device (e.g., smartphone, laptop, etc.) capable of watching videos and able to stream/download videos for viewing and permitting video conferencing

- Reliable internet access

**Inclusion criteria for the infant.**

- Born at term (≥37 weeks' gestation)

- Biological child of the birthing parent

**Exclusion criteria for the birthing parent.**

- Other illicit drug use, besides cannabinoids, during pregnancy (such as heroin and cocaine) per self-report or toxicology results

- Opiate use (prescribed or unprescribed) per self-report or toxicology laboratory results during the immediately completed pregnancy

- Prolonged hospitalization following delivery (>7 days)

**Exclusion criteria for the infant.**

- Major birth defect(s) including physical anomalies including limb malformations/absence of limbs or chromosomal abnormalities

- Diagnosis of neonatal encephalopathy, metabolic disorder, stroke, intracranial hemorrhage, or meningitis during birth hospitalization

- Receipt of any major surgical intervention during the birth hospitalization or prolonged hospitalization (>7 days)

**Sample size determination**

In Objective 1, we anticipate 20 eligible dyads will be approached, consented, and determined to be eligible for study participation, yielding a targeted rate of recruitment of at least 20% of the approached population. A null hypothesis of a rate of 10% will be rejected with 0.89 power using a one-sided exact test at 0.05 significance level.

The intent of Objective 2 is to determine if the retention rate is ≥70%. We anticipate at least 14 dyads will complete the final coaching session for a retention rate of 70% or above. With 20 participants, the null hypothesis that the retention rate is 40% can be tested against the alternative that it is 70% at the one-sided 0.05 significance level with a power of 0.88. With 20 participants, the null hypothesis will be rejected if the number of participants who are retained is ≥14.

The intent of Objective 3 is to determine if the participant adherence rate is ≥80%. We anticipate at least 16 out of 20 birthing parent/infant dyad will complete the required number of INTACT intervention coaching sessions. With 20 participants, the null hypothesis that the adherence rate is 50% can be tested against the alternative that it is 80% at the one-sided 0.05 significance level with a power of 0.90. With 20 participants, the null hypothesis will be rejected if the number of participants who are adherent is ≥16.

## Subject recruitment

Birthing parents with a positive urine toxicology screen for THC or self-reported cannabinoid use during pregnancy will be recruited for participation. The local study team may implement different recruitment options, which could include, but are not restricted to those listed below and in Fig 3. In all aspects of the study, the study personnel will follow local Health Insurance Portability and Accountability Act (HIPAA) regulations and remain mindful of privacy and confidentiality. Study team members will be trained to interact with individuals using a non-judgmental approach. Study personnel interacting with participants have completed institutional implicit bias training, which will aid in a non-judgmental approach.

The enrollment goal for INTACT is 20 birthing parent-infant dyads across three sites within 3 months of site activation. As some sites will have more recruitment barriers, no one site will be allowed to contribute more than half of the total study goal number of dyads (i.e., each site cannot enroll more than 10 dyads). To reach the study enrollment goal over the 3-month period, we expect a monthly enrollment rate of at least 2–3 eligible dyads per site that go on to complete the first coaching session. If the number of participants consented is not reaching the monthly targets, then alternative strategies will be implemented. Additionally, if there are concerns with the recruitment rate, the Data and Safety Monitoring Board (DSMB) will be contacted and notified of the concern, at which time the study team will provide possible solutions and discuss next steps. Study staff will obtain informed consent from the birth parent after delivery. Please see Fig 3 for additional recruitment strategies based on the state cannabinoid legalization status of each study site.

To ensure the study meets the ECHO ISPCTN's goal of providing medically underserved and rural populations with access to state-of-the-art clinical trials, study staff will collect information on each birthing parent-infant dyad regarding diversity, equity, and inclusion. Zip codes will be used to identify individuals from a rural area, defined by a rural-urban community area (RUCA) code of > 4. Individuals will also be asked to self-identify as rural or non-rural. This will be monitored during enrollment with a goal of enrolling 30% of participants are from rural areas.

At the time of recruitment, the NIH Certificate of Confidentiality (CoC) will be discussed with potential participants. The CoC protects information, documents, and/or biospecimens that contain identifiable, sensitive information related to a participant. The CoC policy and 42 U.S. Code 241(d) define identifiable, sensitive information as information that is about an individual and that is gathered or used during the course of research, where an individual is identified or for which there is at least a very minimal risk, that some combination of the information, a request for the information, and other available

| | Long-term Legalization | Recent Legalization | Illegal Status |
|---|---|---|---|
| •MOU agreement to protect against required reporting by providers | | | ✓ |
| •Fliers / direct referrals from clinics | ✓ | ✓ | ✓ |
| •Fliers at local dispensaries | ✓ | ✓ | |
| •Social media outlets | ✓ | ✓ | ✓ |
| •Daily chart review of EMR (includes self-report or positive tox screen) | ✓ | ✓ | ✓ |
| •Approach prenatal participants in ECHO-PASS cohort that self-report THC use | | | ✓ |
| •Use existing collaborations with OB clinics/colleagues | ✓ | ✓ | ✓ |
| •Universal screening of pregnant individuals | ✓ | ✓ | |
| •Approach during birth hospital stay | ✓ | ✓ | ✓ |

**Fig 3. Recruitment strategy among ISPCTN sites with differing legalization status of recreational cannabinoid use.** Created in BioRender. Barber, M. (2025) https://BioRender.com/f698043.

data sources could be used to deduce the identity of an individual. Information that is protected by a CoC is immune from the legal process and is not admissible as evidence unless the participant consents to this disclosure.

For the study site located in a state in which recreational cannabinoid use is illegal, additional measures were completed to ensure participant privacy, and that information is protected from prosecution in the context of a research study. The legal landscape at this site is particularly challenging because illegal substance use during pregnancy falls under the child abuse statute and can lead to civil commitment. Currently, all healthcare providers are mandated to report illegal substance use during pregnancy to the Department of Social Services (DSS) and the Division of Child Protection Services (CPS). To this effect, a Memorandum of Agreement (MOA), a written agreement between researchers and the DSS that serves to honor the protections listed in the CoC, was obtained prior to the start of the study. The MOA allows researchers to collect information about alcohol and drug use during pregnancy from study participants, removes the requirement for research personnel to report this information to DSS, and protects the research personnel from prosecution for withholding such information.

**Outpatient recruitment.** Flyers will be placed in outpatient obstetrical and family medicine clinics affiliated with each of the study sites for birthing parent self-referral. Additionally, providers will be notified of the study enrollment criteria and can distribute a flyer to any pregnant individual who reports cannabinoid use during the 2nd and 3rd trimesters in pregnancy. Study staff will contact potential participants to briefly describe the study and perform a preliminary eligibility screen using an Initial Contact Script. For individuals who indicated interest in the study prenatally, their due date will be noted so that study staff can monitor the hospital patient census for their delivery.

**Inpatient recruitment.** Birthing parents who express interest in the study after seeing a flyer or being referred by healthcare personnel, and who self-reported use or have a positive toxicology screen for THC during the pregnancy will be approached by study staff. The positive toxicology screen may be identified by an automated daily chart review of the electronic medical record (EMR), with the daily report of potentially eligible birthing parents automatically sent via secure email from the EMR to the local study team. A partial HIPAA waiver for this pre-screening procedure to review medical records and identify potential participants will be sought prior to study initiation. Individuals identified in this manner will then be approached by a study team member to introduce the study, gauge interest, and confirm eligibility.

**Community recruitment.** Flyers will be distributed to local cannabinoid dispensaries in the states where recreational use of cannabinoids is legal. For individuals who indicated interest in the study prenatally, their delivery due date will be noted so that study staff can monitor the hospital patient census for their delivery admission.

**Virtual recruitment.** Social media will be considered as a possible recruitment strategy if initial recruitment goals are not being met. Possible strategies for the use of social media will not involve potential participants uploading anything on social media sites that would indicate they may have used cannabinoids during pregnancy.

## Study enrollment and consent

Informed consent will not be obtained until after the birth of the infant to ensure preliminary eligibility. Research coordinators will contact potential participants to briefly describe the study and perform a preliminary eligibility screen. Consent will be obtained within seven days of delivery, followed by completion of the Cannabinoid Use Survey, which ensures the final determination of eligibility to participate in the intervention. Those meeting all eligibility criteria will receive instructions for accessing the ePALS online modules. The participants will provide written informed consent. While the study includes minors, they are below the age of assent. The birthing caregiver provides consent for themselves as well as for the child to participate in the study. There is no waiver from the ethic committee required. The expected dates for the recruitment period are February 2025 – April 2025, with data collection to be completed in May 2026 and data analysis in August 2026.

## Cannabinoid use survey

The final eligibility determination of the birthing parent will be assessed with the Cannabinoid Use Survey, which assesses for multi-substance use of the birthing parent. These data will be collected after the birthing parent consents to study participation. Participants may feel more comfortable providing this information with the added assurance of confidentiality from the CoC and the HIPAA authorization.

The Cannabinoid Use Survey was adapted from the survey being used by the ECHO Cohort component (responsible for observational research with the ECHO Program) and the Early Life Exposures Assessment Tool (ELEAT) [50]. The best measures to identify pregnant people who use cannabinoids during pregnancy is a debated topic in the current literature. Multiple health organizations, including the American College of Obstetricians and Gynecologists (ACOG) and the World Health Organization (WHO), recommend screening for substance use in pregnancy early during prenatal care visits [51]. Universal drug screening is not recommended by any of these groups specifically, and verbal screening (non-structured requests for self-report) is favored by ACOG and the American Society of Addiction Medicine (ASAM) [52]. These recommendations are reflected in the clinical standard of care across the US, in which the majority of substance use in pregnancy is identified by self-report. Even so, there is significant evidence that self-report underestimates the prevalence of substance use, including cannabinoid use [53]. Interestingly, evidence also demonstrates one-time toxicological screening also significantly underestimates cannabinoid use in pregnancy [54]. Historical cohort studies, including Maternal Health Practices and Child Development (MHPCD), the Ottawa Prenatal Prospective Study (OPPS), and Generation R (Gen R) have relied on maternal self-report to identify and quantify cannabis use, all demonstrating differences in neurodevelopmental outcomes of offspring reportedly exposed to cannabinoids in utero using these methods. Gen R obtained urine specimens and self-report measures in a subset of participants and found that 71% of participants with cannabinoid use identified by self-report or toxicology had reported their cannabinoid use to their healthcare provider [54]. Therefore, since self-report has good specificity for identifying cannabinoid use and theoretically may cause fewer feelings of stigmatization and more participant comfort/autonomy, we will use self-report to assess prenatal cannabinoid exposure in this study.

## PALS interventionist/coach certification

Two study coordinators at each site will be certified as a PALS interventionists/coaches to conduct the monthly coaching sessions. The coordinators should have a bachelor's degree. The PALS coach certification process requires active participation in five synchronous remote training sessions with the University of Texas Health Sciences Center at Houston (UTHSCH) and the completion of twelve self-study online modules following the synchronous training sessions. Each synchronous training session lasts two hours and examines the PALS-specific methodology of their birthing parent-infant intervention. The study coordinator will only receive the PALS coach certification after completion of all online modules, synchronous trainings, and passing a knowledge assessment test and practical exam. After coach certification, the study coordinator will be verified to conduct the PALS training methodology and will conduct the monthly coaching sessions at their site with the enrolled birthing parent.

## The INTACT intervention

This study is a behavioral intervention study to assess the feasibility of recruitment, retention, and completion of the intervention in this population. Therefore, no control group and no randomization will occur, and all participants will receive the intervention. The INTACT intervention includes engaging in contingent-response training (i.e., ePALS) by the birthing parent via monthly online modules followed by a personalized coaching session with a PALS-trained coach on a HIPAA-compliant, cloud-based video conferencing service. The INTACT intervention modules are adapted from the ePALS project. The modules are designed to facilitate changes in birthing parents' behaviors across cognitive stimulation

and emotional support domains [48]. Some of the coaching sessions with the birthing parent will be monitored by another interventionist/coach to ensure fidelity throughout the pilot feasibility trial. The consent and HIPAA forms reflect this quality control requirement.

**Study activities and participant timeline**

Following discharge, the birthing parent will complete the following two activities monthly for 12 months: 1) view that month's asynchronous online module, which provides instruction for specified components of contingent parental responses, and 2) complete a synchronous coaching session with their coach, who will provide tailored feedback and advice to the birthing parent regarding the specifics of the online module. The INTACT protocol will require a minimum of two weeks between each coaching session to ensure the intervention continues throughout the study period, rather than clusters of sessions being completed in a short time to ensure steady dosing over the birthing parent's study period.

A maximum of 12 coaching sessions (averaging 1 per month) will occur for the participants. The expected required time devoted to study-related activities for the birthing parent should not surpass 30 hours over the 12-month study period (accounting for consenting, survey completion, and initial basic training). At 12 months of infant age, the number of coaching sessions completed will be collected to describe the intervention implementation accurately (see Fig 4).

**Asynchronous self-guided portions.** Following discharge, the birthing parent's coach will invite (i.e., electronically release) the birthing parent to complete the first ePALS module. They will become oriented to the INTACT intervention and the ePALS platform with this first ePALS module and will complete the first module during their infant's first month of life. Within three weeks of their discharge, the birthing parent will be contacted via telephone (and/or their preferred mode of communication) by their coach to schedule their first coaching session. Questions and/or concerns will be addressed during this communication.

The birthing parent will complete an assigned asynchronous INTACT intervention module each month (total of 12 modules) before their scheduled synchronous coaching session. The online modules are up to 60 minutes in length and can be completed at leisure (i.e., not required to be completed in one sitting). The birthing parent will interact with the program

## Legal Status of Cannabis use in IDeA States

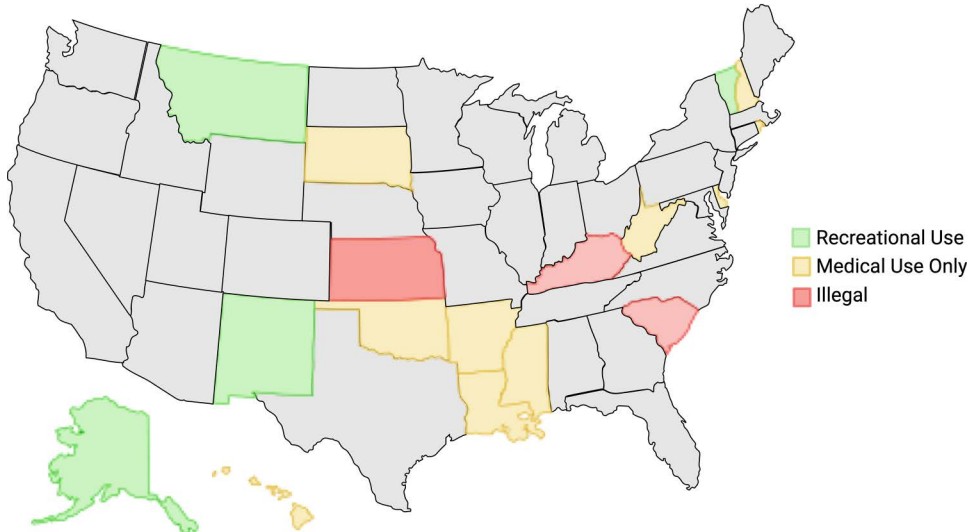

**Fig 4. Enrollment and study activities.**

by watching narrated video segments and answering questions (e.g., open-ended; multiple-choice) about the program's content and about their own interactions with their infant. Their coach will be able to monitor for completion of the module throughout the month and provide reminders as that month's coaching session approaches.

The skills taught in the modules include reading infant signals, responding with warm and sensitive behaviors, maintaining versus redirecting infants' focus of attention, watching for opportunities to introduce an object or social game, stimulating language development by naming objects and action in combination with physical demonstrations, and incorporating the use of the constellation of behaviors during daily tasks such as dressing and feeding. Not all modules follow the same format. For instance, one online module is a review session designed to consolidate the learning and the opportunity to demonstrate their new skills to another adult who participates in the baby's life.

**Synchronous coaching sessions.** Monthly coaching sessions between the birthing parent and their coach will occur via a HIPAA-compliant cloud-based video conferencing service, lasting no more than 60 minutes. The session will begin with reflecting on the progress, or any setbacks experienced in the previous month, and will then focus on the contingent parenting techniques and feedback from the most recent online module. Specific components from the online module will be discussed alongside the reiteration and reinforcement of birthing parent-infant trust, responsiveness, attachment, acceptance, accountability, and reflection. During sessions, the birthing parent participant may interact with the infant to model or practice behaviors with the coach observing to provide immediate feedback (i.e., positive reinforcement and suggestions for improvement). At the close of the call the coach will reinforce the topics discussed in the call and will confirm with the birthing parent a goal (or a few) to focus on until the next scheduled coaching session (e.g., talking with their infant throughout bath time; maintaining their attention during play). The following coaching session will be scheduled before the close of the call. Following the call, the coach will invite (i.e., electronically release) the birthing parent to complete the following asynchronous module.

## Participant retention

As one of the primary outcomes of this pilot trial is the retention of the birthing parent-infant dyad throughout all study activities, particular attention will be given to different retention strategies. We anticipate up to 30% of participants not completing all study activities. This may occur for a variety of reasons, such as unanticipated stress of a new infant at home, balancing their work schedule with a new infant, disinterest or boredom while completing the online modules and coaching sessions, and/or discomfort in the vulnerability of undergoing parental coaching. The study team will make all reasonable attempts to keep the birthing parent active in the study and will accommodate their life circumstances as able within protocol requirements.

**Reminders and scheduling assistance.** The study team will periodically communicate with the birthing parent throughout the month, providing reminders of activities and upcoming coaching sessions and showing appreciation for the completion of activities. If a birthing parent does not complete either monthly task (asynchronous training and synchronous coaching session), the study team will contact them via their preferred contact method to determine if the birthing parent would like to continue in the study. If they express continued interest, the study team will provide a reminder of the outstanding tasks and assist with any technical or scheduling difficulties. To accommodate the need to reschedule a coaching session, there will be flexibility of ± 7 days to complete a monthly coaching session. Additionally, birthing parents are allowed 3 consecutive missed coaching sessions before withdrawal from the study.

To increase retention, at the close of each coaching session the coach will schedule the following coaching session with the birthing parent. This will ensure that there is a tangible plan and deadline for the birthing parent to complete their asynchronous module.

**Participation stipend.** Participants will be reimbursed monthly for their time and effort in participating in the study. Each month the participant will receive compensation ($20) for completing the coaching session. An internet stipend ($20) will be provided for months 1–12 to subsidize internet access to complete study activities. Additionally, compensation will be provided upon completion of the pre-intervention study tasks during the birth hospitalization.

## Coaching reliability and fidelity

To ensure reliability and fidelity across coaches and birthing parents, the same coach will complete all coaching sessions for a given birthing parent if possible. If a coach is unable to complete a coaching session, the other coach at the site can assist with the session, however, the original coach will step back in for subsequent sessions. All efforts will be made by the site to minimize switching of coaches for one participant. If a coach leaves the study, the other coach at the study site may complete any remaining coaching sessions with the participant. The study team hopes to build trust and repertoire between the birthing parent and the coach as much as possible, rather than introducing a different coach or swapping coaches between months, which could introduce discontinuity in the coaching strategy for the birthing parent.

Furthermore, early parenting interventions can hinge on the establishment of trust between the birthing parent and coach, as initiating and nurturing behavioral change can be difficult for some individuals [55]. The coach must be emotionally and culturally sensitive towards the birthing parent. By actively acknowledging the inherent challenges of change within the context of new parenting and demonstrating empathy and encouragement, the coach can increase retention by cultivating a sense of self-efficacy and protocol adherence.

## Data management

The data management plan is consistent with those described in the Good Clinical Data Management Practices. The trial staff at the individual sites are responsible for data collection under the supervision of the site investigator, who is responsible for ensuring the accuracy, completeness, legibility, and timeliness of the data reported. Each IRB-approved site entering data will perform internal quality management of study conduct, data collection, documentation, and completion. It is best practice for site coordinators to use hardcopies of any data recorded on paper case-report forms or trial visit worksheets/assessment forms as source document worksheets for recording data for each participant consented. Data recorded in the electronic data capture system derived from source documents must be consistent with the data recorded on the source documents. Site personnel will enter data (including demographics) into the electronic data capture system that complies with the Health Insurance Portability and Accountability Act (HIPAA) regulations. The electronic data capture system includes password protection and internal quality checks, such as automatic range checks, to identify data that appear inconsistent, incomplete, or inaccurate. Data will be de-identified before sharing externally.

Sites will provide direct access to all their facilities, source data/ documents, and reports for the purpose of monitoring and auditing by the Data Coordinating and Operations Center and inspection by local and regulatory authorities. The monitors will verify that the clinical trial is conducted, and that data is generated, documented (recorded), and reported in compliance with the protocol, the trial-specific site performance plan, site-specific standard operating procedures, the Good Clinical Practice E6(R2), and applicable regulatory requirements. The Data Coordinating and Operations Center will implement quality control procedures for the database and records in accordance with the site performance plan, manual of procedures, data safety monitoring plan, and applicable standard operating procedures. The Data Coordinating and Operations Center will address issues uncovered during quality assurance, quality control, or monitoring activities through simple corrections or root-cause analysis, followed by instituting corrective and preventative action, as appropriate.

## Statistical analyses

Descriptive statistics will be summarized for demographics and cannabinoid usage collected at baseline. All numerical variables will be summarized using mean ± standard deviation and median (minimum, maximum). All categorical variables will be summarized using frequency (in %). The sample size is based on practical considerations such as the estimated number of potential participants from the sites and the timeline of the study. Considering the small sample size proposed in the pilot study, exact methods will be used in statistical inference. Hypothesis tests will be one-sided with a significance level of 0.05.

For each objective, quantitative benchmarks for feasibility measures have been set by defining the targeted rates for recruitment, retention, and adherence. The first objective, participant recruitment, will be summarized using frequency (in %). A one-sided exact Clopper-Pearson (or exact) test will be used to test against a null rate of 10%. Objective two, participant completion, will be summarized using frequency (in %) and a one-sided exact Clopper-Pearson (or exact) test will be used to test against a null rate of 40%. The third objective, participant adherence, will be summarized using frequency (in %) and a one-sided exact Clopper-Pearson test will be used to test against a null rate of 50%.

### Participant safety

The data safety monitoring board (DSMB) has reviewed and approved the protocol prior to implementation. Interim monitoring by the DSMB will occur of accumulating data from research activities to assure the continuing safety of research participants, relevance of the study questions, and appropriateness of the study and integrity of the accumulating data. The study progress review plan includes a monthly meeting with the site principal investigators, research coordinators, and mentoring faculty to review the rate of enrollment (when applicable) and the retention and completion of the intervention. The purpose of these meetings will be to ensure that targets are being meet and to discuss any concerns with the study. If concerns arise, or the targets are not being met, then the DSMB will be contacted. Additionally, all reported adverse events will also be reviewed along with the current literature to ensure that no change to the study or protocol is indicated.

## Discussion

### Anticipated recruitment hurdles

This protocol purposefully includes sites across the US with varying stages of the legality of recreational cannabinoid use to assess feasibility of recruitment. Hurdles that may impact recruitment and participation may include 1) personal stigma surrounding cannabis, 2) fear of legal repercussion, and 3) limitations on recruitment methods.

Worry of personal or communal stigma may prevent individuals from volunteering to participate in research should they admit to past or current use of cannabinoids. Even in the context of a confidential and compliant study, there may be worry about professional (e.g., fear of losing a job) or personal repercussions (e.g., strain on familial relationships) if their use should become known to others.

Additionally, individuals may fear legal repercussions, especially if they have had past experiences with the law. Regarding the involvement of children, birthing parents may fear the involvement of child protective services (CPS), even in states where it remains legal for recreational use [56]. This is most likely because cannabis remains a Schedule 1 drug federally and can create confusion as CPS is held to comply with both federal and state law. As states are recently legalizing the recreational use of cannabinoids, there is a policy gap between federal and state rulings that have yet to be closed. The study team worked extensively with the central and local IRBs and legal teams to frame the consent documentation and other study documents to address these concerns as transparently and accurately as possible for each site.

Finally, recruitment methods may be limited in some states, as advertising regarding an illegal substance may be restricted; this may limit recruitment. These potential hurdles highlight the critical need for this type of study to establish the feasibility and identify possible solutions to recruitment hurdles in this vulnerable population, so that we can safely and rigorously assess the impact of cannabinoid use on child health.

## Supporting information

**S1 File:  SPIRIT checklist.** The SPIRIT checklist for the INTACT study.
(DOCX)

**S2 File: Protocol.** The INTACT study protocol.
(DOCX)

## Acknowledgments

The authors would like to acknowledge the contributions of Dr. Susan Landry and the PALS team at The Children's Learning Institute at UTHealth Houston.

## Author contributions

**Conceptualization:** Jessie Maxwell, Leigh-Anne Cioffredi, Maria M. Talavera-Barber, Laurie Chassereau, Hengameh Raissy.

**Data curation:** Sandra Beauman, Anne Hittson, Meggie McCoy.

**Formal analysis:** Jing Jin.

**Funding acquisition:** Hengameh Raissy.

**Investigation:** Jessie Maxwell, Matthew Henry, Preetha A. Abraham, Linda Y. Fu.

**Methodology:** Jessie Maxwell, Leigh-Anne Cioffredi, Maria M. Talavera-Barber, Matthew Henry, Sandra Beauman, Anne Hittson, Meggie McCoy, Laurie Chassereau, Jing Jin, Preetha A. Abraham, Linda Y. Fu, Hengameh Raissy, Jessica N. Snowden.

**Project administration:** Jessie Maxwell, Matthew Henry, Sandra Beauman, Anne Hittson, Meggie McCoy, Laurie Chassereau, Preetha A. Abraham, Linda Y. Fu.

**Resources:** Preetha A. Abraham, Linda Y. Fu, Hengameh Raissy.

**Supervision:** Jessie Maxwell, Preetha A. Abraham, Linda Y. Fu, Hengameh Raissy, Jessica N. Snowden.

**Writing – original draft:** Jessie Maxwell, Leigh-Anne Cioffredi, Maria M. Talavera-Barber, Jing Jin, Preetha A. Abraham, Linda Y. Fu.

**Writing – review & editing:** Jessie Maxwell, Leigh-Anne Cioffredi, Maria M. Talavera-Barber, Matthew Henry, Sandra Beauman, Anne Hittson, Meggie McCoy, Laurie Chassereau, Jing Jin, Preetha A. Abraham, Linda Y. Fu, Hengameh Raissy, Jessica N. Snowden.

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
