## [Decision Letter · Decision Letter 0]

18 Feb 2025

PONE-D-24-53860Pilot Feasibility Trial of Improving Neurodevelopmental ouTcomes After prenatal Cannabinoid in uTero exposure (INTACT) Study Protocol for a Multi-Center TrialPLOS ONE

Dear Dr. Maxwell,

Thank you for submitting your manuscript to PLOS ONE. After careful consideration, we feel that it has merit but does not fully meet PLOS ONE’s publication criteria as it currently stands. Therefore, we invite you to submit a revised version of the manuscript that addresses the points raised during the review process.

We look forward to receiving your revised manuscript.

Kind regards,

Andrea Mastinu

Academic Editor

PLOS ONE

 [The Environmental influences on Child Health Outcomes (ECHO) Program, Office of the Director, National Institutes of Health, supported this research under the award numbers U24OD024957, UG1OD024947, UG1OD024955, and UG1OD030019.]. 

5.  Please include a caption for figure 1, 2, 3 and 4.

6. Upon checking, your supplementary figures is same in your figures. Please remove the duplicate figures.

Reviewers' comments:

Reviewer's Responses to Questions

**Comments to the Author**

1. Does the manuscript provide a valid rationale for the proposed study, with clearly identified and justified research questions?

Reviewer #1: Yes

2. Is the protocol technically sound and planned in a manner that will lead to a meaningful outcome and allow testing the stated hypotheses?

Reviewer #1: Yes

3. Is the methodology feasible and described in sufficient detail to allow the work to be replicable?

Reviewer #1: No

4. Have the authors described where all data underlying the findings will be made available when the study is complete?

Reviewer #1: No

5. Is the manuscript presented in an intelligible fashion and written in standard English?

Reviewer #1: Yes

6. Review Comments to the Author

You may also provide optional suggestions and comments to authors that they might find helpful in planning their study.

Reviewer #1: This is a protocol to conduct a feasibility pilot study of a novel program (not clinical intervention) to improve neurodevelopmental outcomes in infants prenatally exposed to cannabinoids as the basis for a future randomized controlled trial. The investigators will conduct the feasibility trial in three ISPCTN sites to assess the following variables: participant recruitment, participant completion when the infant is 12 months of age, and completion of overall total number of INTACT intervention coaching sessions. This information will inform a future randomized controlled trial. Thus, no clinical outcome assessment will be completed in this feasibility pilot study.

Two states (Vermont, New Mexico) having legal use and one state (South Dakota) having legal medical use will participate. It appears that this research group gave much thought and consideration for understanding and negotiating the challenges of the cannabis environment and subject protection.

The recruitment and retention strategies are well outlined. (outpatient, inpatient, community, virtual). The 20 dyads is the basic sample size which is small as expected in this pilot setting. The interventions (INTACT coaching and PALS training) are all well planned .

The statistics for analysis are exact methods due to the small sample size. So the exact approach such as the Clopper Pearson method is reasonable for the hypothesis for testing the adequacy of the samples in the three objectives. The Clopper Pearson approach, which is exact and routine, for this aspect will suffice. The statistical analysis section is brief and descriptive as expected for this feasibility effort. Presumably for the success of the INTACT methodology and other endpoints in a future comparative trial the investigators are considering a more involved inferential approach to the analysis. Section 8.1 in the included protocol draft includes mainly comparison of proportions. Hopefully , given a larger study, some more sophisticated multivariate models or general linear models approaches will be entertained.

The data management section of this pilot is rather brief. It needs an expansion of quality control methodology for data collection in the field as well as for central data processing in the main database.

7. PLOS authors have the option to publish the peer review history of their article (what does this mean? ). If published, this will include your full peer review and any attached files.

**Do you want your identity to be public for this peer review?** For information about this choice, including consent withdrawal, please see our Privacy Policy .

Reviewer #1: No

---

## [Author Response · Author response to Decision Letter 0]

28 Feb 2025

To Whom It May Concern:

We are grateful for the feedback received and the opportunity to improve the manuscript submission. Please see our responses below to each of the comments provided. We look forward to your response after the revisions have been reviewed.

Editor Comments:

1. Please ensure that your manuscript meets PLOS ONE’s style requirements, including those for file naming.

We have revised the formatting of the manuscript per the PLOS Study Protocol Articl Template. This included moving the author contributions immediately after the discussion, followed by acknowledgements, supporting information which includes abbreviations, and references. The author contributions were revised to use the CRediT taxonomy descriptions. The availability of the data and materials was updated. The abstract was shortened to meet the 300-word limit.

2. Thank you for stating the following financial disclosure. Please state what role the funders took in the study. If the funders had no role, please state: The funders had no role in study design, data collection and analysis, decision to publish, or preparation of the manuscript.

We apologize for the oversight in not originally providing these details. The financial disclosure statement has been revised and now includes the following: “As this study is being conducted under cooperative agreement funding mechanisms, LYF and PA, employees of the National Institute of Health, participated in study design, preparation of the manuscript and the decision to publish. The content is solely the responsibility of the authors and does not represent the official views of the National Institute of Health.” This information has also been added to the cover letter as recommended.

3. Please amend either the title on the online submission form or the title in the manuscript so that they are identical.

This oversight has been resolved, with the titles now identical between the online submission form and in the manuscript.

4. Your ethics statement should only appear in the Methods section of your manuscript.

The ethics statement has been moved from the end of the manuscript to the methods section as suggested.

5. Please include a caption for Figure 1, 2, 3 and 4.

The captions for the Figures have been added to the text as recommended in the guidelines for submission. We apologize for not originally including this information.

6. Upon checking, your supplementary figure is the same in your figure. Please remove the duplicate figures.

The supplementary figures that were previously included are the licenses allowing publication that are from BioRender, where the figures were created. The title of these files has been clarified (example - Fig 1 License) to minimize confusion.

7. Please review your reference list to ensure that it is complete and correct.

The reference list has been reviewed to ensure completeness and that it is correct.

Reviewer Comments:

1. Is the methodology feasible and described in sufficient detail to allow the work to be replicated? No

Additional information has been added to the data management section and the statistical analysis section, which should now provide sufficient detail to allow the work to be replicated. Regarding the statistical analysis, the primary goal of this pilot study is to assess the feasibility of the intervention. A single-arm design allows the study team to focus on this aspect without the complexity of a control group. Also, a single-arm design can be more practical and cost-effective. Pilot studies are not typically designed for formal hypothesis testing. A precise sample size calculation is not necessary. In the study protocol, we proposed the sample size based on practical considerations such as the estimated number of potential participants from the sites and the timeline of the study. For each objective, we set quantitative benchmarks for feasibility measures by defining the targeted rates for recruitment, retention, and adherence.

2. Have the authors described where all data underlying the findings will be made available when the study is complete? No

We apologize for this oversight and have revised the availability of data and materials. This now reads “In accordance with the NIH Data Management Sharing Policy, the de-identified dataset for this trial will be uploaded at the time of data publication to the NICHD Data and Specimen Hub (DASH), a controlled-access, public use database.”

3. The data management section of this pilot is rather brief. It needs an expansion of quality control methodology for data collection in the field as well as for central data processing in the main database.

Per the recommendation, we have expanded the quality control methodology for data collection in the field as well as for central data processing in the main database. We believe that the added information strengthens this section and now meets reviewer expectations.

---

## [Decision Letter · Decision Letter 1]

16 Mar 2025

The Protocol for a Pilot Feasibility Trial of Improving Neurodevelopmental ouTcomes After prenatal Cannabinoid in uTero exposure (INTACT) Study for a Multi-Center Trial

PONE-D-24-53860R1

Dear Dr. Maxwell,

We’re pleased to inform you that your manuscript has been judged scientifically suitable for publication and will be formally accepted for publication once it meets all outstanding technical requirements.

Kind regards,

Andrea Mastinu

Academic Editor

PLOS ONE

Additional Editor Comments (optional):

Reviewers' comments:

Reviewer's Responses to Questions

**Comments to the Author**

1. Does the manuscript provide a valid rationale for the proposed study, with clearly identified and justified research questions?

Reviewer #1: Yes

2. Is the protocol technically sound and planned in a manner that will lead to a meaningful outcome and allow testing the stated hypotheses?

Reviewer #1: Yes

3. Is the methodology feasible and described in sufficient detail to allow the work to be replicable?

Reviewer #1: Yes

4. Have the authors described where all data underlying the findings will be made available when the study is complete?

Reviewer #1: Yes

5. Is the manuscript presented in an intelligible fashion and written in standard English?

Reviewer #1: Yes

6. Review Comments to the Author

You may also provide optional suggestions and comments to authors that they might find helpful in planning their study.

Reviewer #1: No Comment.

XXXXXXXXXXXXXXXXXXXXXXXXXXXXXXXXXXXXXXXXXXXXXXXXXXXXXXXXXXXXXXXXXXXXXXXXXXXXXXXXXXXXXXXXXXXXXXXXXXXXXXXXXXXXXX.

7. PLOS authors have the option to publish the peer review history of their article (what does this mean? ). If published, this will include your full peer review and any attached files.

**Do you want your identity to be public for this peer review?** For information about this choice, including consent withdrawal, please see our Privacy Policy .

Reviewer #1: No

---

## [Editor Report · Acceptance letter]

PONE-D-24-53860R1

PLOS ONE

Dear Dr. Maxwell,

I'm pleased to inform you that your manuscript has been deemed suitable for publication in PLOS ONE. Congratulations! Your manuscript is now being handed over to our production team.

Kind regards,

on behalf of

Dr. Andrea Mastinu

Academic Editor

PLOS ONE